# Codon choice directs constitutive mRNA levels in trypanosomes

Janaina de Freitas Nascimento[1†], Steven Kelly[2†], Jack Sunter[1‡], Mark Carrington[1*]

[1]Department of Biochemistry, University of Cambridge, Cambridge, United Kingdom; [2]Department of Plant Sciences, University of Oxford, Oxford, United Kingdom

**Abstract** Selective transcription of individual protein coding genes does not occur in trypanosomes and the cellular copy number of each mRNA must be determined post-transcriptionally. Here, we provide evidence that codon choice directs the levels of constitutively expressed mRNAs. First, a novel codon usage metric, the gene expression codon adaptation index (geCAI), was developed that maximised the relationship between codon choice and the measured abundance for a transcriptome. Second, geCAI predictions of mRNA levels were tested using differently coded GFP transgenes and were successful over a 25-fold range, similar to the variation in endogenous mRNAs. Third, translation was necessary for the accelerated mRNA turnover resulting from codon choice. Thus, in trypanosomes, the information determining the levels of most mRNAs resides in the open reading frame and translation is required to access this information.
DOI: https://doi.org/10.7554/eLife.32467.001

*For correspondence:
mc115@cam.ac.uk

[†]These authors contributed equally to this work

Present address: [‡]Department of Biological and Medical Sciences, Oxford Brookes University, Oxford, United Kingdom

Competing interests: The authors declare that no competing interests exist.

## Introduction

In trypanosomatids, RNA polymerase II dependent transcription of protein coding genes is polycistronic (*Van der Ploeg, 1986*; *Worthey et al., 2003*). Individual monocistronic mRNAs are produced after co-transcriptional processing by *trans*-splicing a 39 nucleotide, capped, exon to the 5' end and linked endonucleolytic cleavage and polyadenylation of the upstream mRNA (*Sutton and Boothroyd, 1986*; *Murphy et al., 1986*; *Ullu et al., 1993*; *Matthews et al., 1994*). This mechanism of mRNA transcription has co-evolved with the structure of the genome; protein coding genes occur in tandemly arrayed clusters containing up to hundreds of genes in one transcription unit (*Ivens et al., 2005*; *Berriman et al., 2005*). The transcription start sites for these gene clusters have been identified (*Kolev et al., 2010*) and are marked by histone variants (*Siegel et al., 2009*) and although the nature of promoters remains incompletely defined, a GT-rich promoter element has been identified that can both initiate transcription and cause histone variant deposition (*Wedel et al., 2017*). There is remarkable synteny between the genomes of trypanosomatid species that diverged hundreds of millions of years ago (*El-Sayed et al., 2005*) suggesting a functional organisation of genes within polycistronic transcription units. In most cases, this is not due to the presence of an equivalent of prokaryotic operons but appears to be linked to the distance from the transcription start site. For example, genes up-regulated during heat shock tend to be distal to the transcription start site so that they continue to be transcribed by elongating RNA polymerase II for >60 min after transcription initiation has stopped as a consequence of the stress (*Kelly et al., 2012*). As yet, there is no convincing evidence for selective regulation of RNA polymerase II transcription of individual genes or transcription units.

In the absence of regulation based on selective transcription of individual genes, the disparate levels of individual mRNAs within a trypanosome cell is dependent solely on post-transcriptional processes. In trypanosomatids, some very abundant RNA polymerase II transcribed mRNAs are encoded

**eLife digest** Genes are made up of DNA and contain the instructions to make molecules called proteins. This information is stored as a genetic code consisting of four bases: adenine (A), cytosine (C), guanine (G) and thymine (T). The order of these bases and their different combinations serves as a blueprint for making thousands of different proteins and to assemble living cells.

Converting the information in the genes into proteins requires several steps. First, the code from the DNA needs to be transcribed into RNA and then processed to make messenger RNA, or mRNA for short, which in turn is translated into proteins. Cells decode mRNAs by reading the bases as groups of three, also called codons. Most codons specify an amino acid – the building blocks of proteins – but certain codons also mark the start and end point of a protein. This ensures that the mRNA is read in the correct 'frame' and the desired proteins are made.

Any cell contains thousands of different proteins and each protein has its own unique level. The mechanisms used to set the number of each different type of protein can operate at every point in the process. In many organisms, the number of times a gene is transcribed to make an mRNA, underpins differences in protein levels. Trypanosomes, for example, are parasites that cause a range of devastating diseases in humans and livestock. They lack the ability to set individual mRNA levels by regulating how often the gene is transcribed. This suggests that the expression of thousands of mRNAs is regulated by a common control mechanism later in the process ending in protein synthesis. However, until now, it was unclear what these mechanisms are.

Most amino acids are encoded by more than one codon. The different codons for one amino acid are not equivalent and using a different codon can lead the mRNA to yield more or less protein. Evolution acts on these differences between codons, and the 'codon choice' in any one mRNA represents the outcome of natural selection. Now, Nascimento, Kelly et al. found that codon choice directs both the levels of mRNAs and the level of translation.

For the experiments, a new metric that enables a prediction of the level of expression for each mRNA was created. This metric (known as the 'gene expression codon adaptation index' or geCAI for short) could relate the codon choice to mRNA levels. For example, mRNAs with a low index score had shorter half-lives, i.e., how long that mRNA remained in the cell before being broken down.

Nascimento, Kelly et al. confirmed this by measuring mRNA levels in specific genes tagged with distinguishable markers and revealed that the codon choice indeed dictated the rate at which an mRNA would be broken down. A separate study by Jeacock, Faria and Horn looked more closely at how codon choice contributes to the control of the copy number of proteins. However, genes and mRNAs involved in development could deviate from the levels predicted by the geCAI metric, which suggests that other mechanisms may be in place to control the stability these mRNAs.

The importance of codon choice in setting mRNA levels has now been demonstrated in several organisms, including yeast and trypanosomes, which suggests that this process is more widespread than previously realised.

DOI: https://doi.org/10.7554/eLife.32467.002

by tandemly arrayed multigene families so one pass of the RNA polymerase transcribes multiple copies of the genes in series producing increased numbers of pre-mRNAs. For example, α- and β-tubulins are encoded by a tandem array of 19 copies of each gene (*Ersfeld et al., 1998*). However, the majority of genes are single copy and are expressed in the cell at different levels (*Kolev et al., 2010*) so mechanisms for regulation of mRNA abundance must exist and these must act post-transcriptionally.

It is probable that the abundance of each mRNA is regulated at several levels during its production/degradation cycle: the rate of maturation against the decay rate of the pre-mRNA in the nucleus; the rate of export to the cytoplasm against the half-life of the mature mRNA in the nucleus; the half-life of the mature mRNA in the cytoplasm (*Haanstra et al., 2008*). Most work aimed at understanding determinants of mRNA levels has investigated *cis*- and *trans*-acting factors that modulate the cytoplasmic half-life of the minority of mRNAs that alter in response to a developmental or an environmental trigger. In *Trypanosoma brucei* there are a series of developmental forms that are

adapted to different host niches and the variation in mRNA expression levels between the mammalian bloodstream form (BSF) and the insect midgut procyclic form (PCF) has been used to assay for factors necessary for a decreased half-life in one developmental form. *Cis*-elements have been identified in the 3'UTRs that differentially affect stability and/or rate of translation of mRNAs in specific developmental stages; the best characterised being those in procyclin mRNAs (*Furger et al., 1997*; *Hehl et al., 1994*), cytochrome oxidase subunits (*Mayho et al., 2006*), and a short stem loop, necessary and sufficient for response to external purine concentration (*Fernández-Moya et al., 2014*). Manipulation of these elements rarely quantitatively recapitulates the observed variations in mRNA levels in vivo whereas transfer of an entire 3'UTR can (*Webb et al., 2005a*), this may reflect a combinatorial mechanism necessary for complete developmental regulation.

Trans-acting factors that affect mRNA levels have been identified. The most spectacular examples of these include the RNA recognition motif-containing proteins RPB6 and RPB10. Over expression of RBP6 is sufficient to drive PCFs through several successive developmental transitions (*Kolev et al., 2012*) and over expression of RBP10 causes a PCF transcriptome to switch to BSF (*Mugo and Clayton, 2017*). Other RNA binding proteins have been shown to have specific effects on sets of mRNAs, for example RBP42 binds within the open reading frame (ORF) of many mRNAs that encode proteins involved in energy metabolism (*Das et al., 2012*). Similarly, ZPF3 binds the LII *cis*-element in the EP1 procyclin mRNA 3' UTR and displaces a negative regulator of translation leading to increased EP1 protein abundance (*Walrad et al., 2009*; *Walrad et al., 2012*).

These observations provide strong evidence for co-ordinated regulation of mRNA cohorts and have led to a model in which the fate of cytoplasmic mRNAs is regulated by a combinatorial mechanism dependent on a set of interactions between RNA-binding proteins and their cognate sites in individual mRNAs. In this model, the information for the range of possible fates for any mRNA is contained in discrete *cis*-elements. However, the model is derived from a consideration of a small subset of mRNAs prone to dramatic regulation and further, it fails to consider work that demonstrated a codon bias in highly expressed genes in *T. brucei* and related trypanosomatid species (*Horn, 2008*; *Seward and Kelly, 2016*). The use of favoured codons in highly expressed genes correlated with cognate tRNA gene numbers and led to the suggestion that translational selection was likely to act on highly abundant genes. Moreover, selection was also acting in proportion to mRNA abundance to reduce the biosynthesis requirements of those transcripts (*Seward and Kelly, 2016*). More recently, it has been shown that codon choice is a major determinant of mRNA levels in yeast (*Presnyak et al., 2015*) operating through a system that monitors the rate of translation elongation involving displacement of the RNA helicase DHH1 which in turn affects the activity of the NOT1 deadenylase complex (*Radhakrishnan et al., 2016*). Further, codon choice is also an important determinant of maternal mRNA turnover in zebra fish (*Mishima et al., 2016*) and Drosophila (*Bazzini et al., 2016*).

Here, we provide evidence that codon use is also central to the regulation of mRNA levels in trypanosomes. A novel codon usage statistic, the gene expression codon adaptation index (geCAI), was developed that maximises the relationship between codon usage and the measured transcript abundance for the full range of mRNAs present in a cell. Then geCAI predictions were tested using differently coded GFPs and mRNA levels were predicted with >90% accuracy over a ~ 25 fold dynamic range. An investigation of the mechanism of how geCAI determined mRNA levels showed that mRNAs with low geCAI scores had shorter half-lives and that turnover was prevented if translation was blocked. These observations and measurements show that for most mRNAs the majority of the information determining mRNA levels resides in the ORF and is accessed by the translating ribosome.

## Results

The experiments described below originated from an observation made during attempts to reduce expression of a transgene encoding a MS2 binding protein-green fluorescent protein-nuclear localisation signal (MS2bp-GFP-nls) fusion protein. Of the approaches used, adjusting the open reading frame (ORF) to codons that were under-represented in highly expressed mRNAs was more effective than other modifications including mutating the AG acceptor dinucleotide to AA in the trans-splicing site (*Siegel et al., 2005*), using a synthetic 5'UTR (*Siegel et al., 2005*) and using a 3'UTR from a low abundance mRNA (*Supplementary file 1*). The relative levels of the MS2bp-GFP-nls mRNA were

then estimated by northern blotting (*Figure 1*), this showed a correlation between levels of MS2bp-GFP-nls mRNA and protein (*Figure 1—figure supplement 1*) levels, including the codon altered transgene mRNA. This observation implied that codon choice could determine mRNA levels and the relationship between the two was investigated further.

## Altering codon use results in a range of GFP protein expression levels

All subsequent experiments used procyclic form trypanosomes and GFP as a reporter. A range of GFP transgenes were integrated into the tubulin locus and were expressed after endogenous transcription by RNA polymerase II (*Figure 2*). The transgene-derived GFP mRNA contained an α-tubulin 5'UTR, the GFP ORF and the actin 3'UTR (*Supplementary file 2A*). Different GFP ORFs were selected from a pre-existing library (*Kudla et al., 2009*) and others were newly synthesised (*Supplementary file 3*). On analysing transgenic cell lines, it was immediately apparent that different GFP ORFs produced a range of fluorescence levels (*Figure 2—figure supplement 1*).

Prior to further experiments, the use of fluorescence as a proxy for GFP protein was investigated by comparing GFP expression levels measured using flow cytometry with estimates from western blotting. Three independent clones of cell lines expressing GFPs with different codon use (eGFP, GFP 065, GFP 226 and GFP 102; *Supplementary file 3*) were used. GFP levels were measured by flow cytometry of live cells and lysates from the same cultures were used to estimate GFP expression using densitometric scanning of a western blot and comparison with a standard curve of recombinant GFP (*Figure 2—figure supplement 2*). The Pearson's correlation coefficient between the

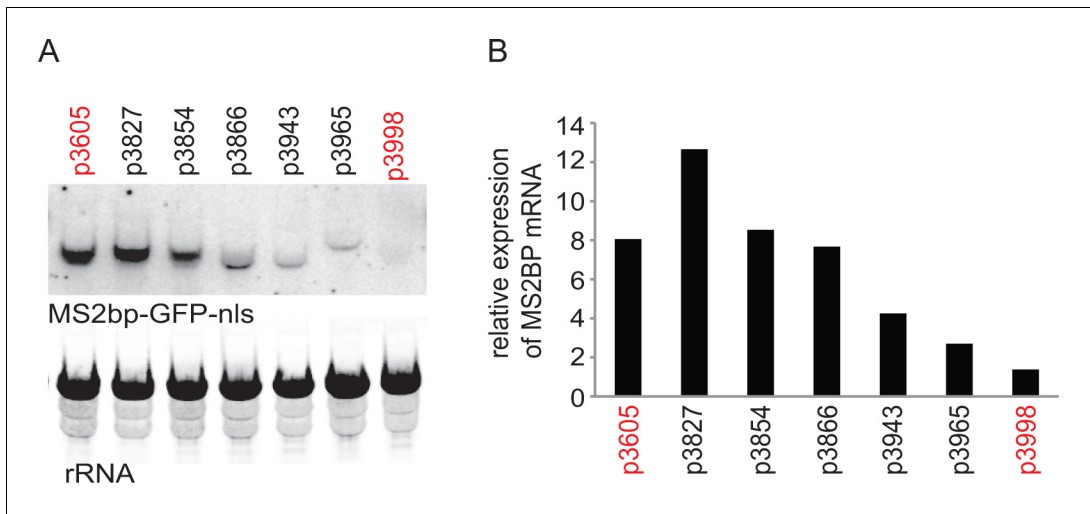

**Figure 1.** Initial experiment that provided evidence that altering codon use of a transgene ORF affected mRNA as well as protein levels. A series of MS2bp-GFP-nls transgenes were integrated into the tubulin locus of procyclic form cells and expressed by endogenous transcription. Starting with p3605 which directed the expression of MS2bp-GFP-nls with α-tubulin 5'UTR and actin 3'UTR the following changes were made: p3827, substituted β-tubulin 5'UTR; p3854, substituted synthetic 5'UTR (based on pNS11 from [*Siegel et al., 2005*]); p3866, as p3854 but *trans*-slice acceptor site mutated from AG to AA; p3943, as p3866 but sequence around initiation codon changed from CCGCCGCCATG to TTTTTTTATG p3965, as p3943 but substituted RAB28 3'UTR p3998, as p3605 but with the ORF altered to contain codons enriched in genes with low expression levels. All sequences are in *Supplementary file 1*. (A) The relative expression of the transgene mRNAs was estimated by northern blotting of total RNA and detected using a probe made from an equal mixture of 'wild type' and re-codoned MS2bp-GFP-nls ORF sequences; the probe was in excess. (B) The relative expression of transgene mRNAs was estimated by phosphorimaging and adjustment for loading using rRNA signal. The effect of the use of infrequent codons can be seen by comparing p3605 and p3998, in red.

DOI: https://doi.org/10.7554/eLife.32467.003

The following figure supplement is available for figure 1:

**Figure supplement 1.** Diagram showing GFP fluorescence detected by flow cytometry in 20 000 cells for each transgene expressing cell line and the parental *T. brucei* Lister 427 KG procyclic form cell line.

DOI: https://doi.org/10.7554/eLife.32467.004

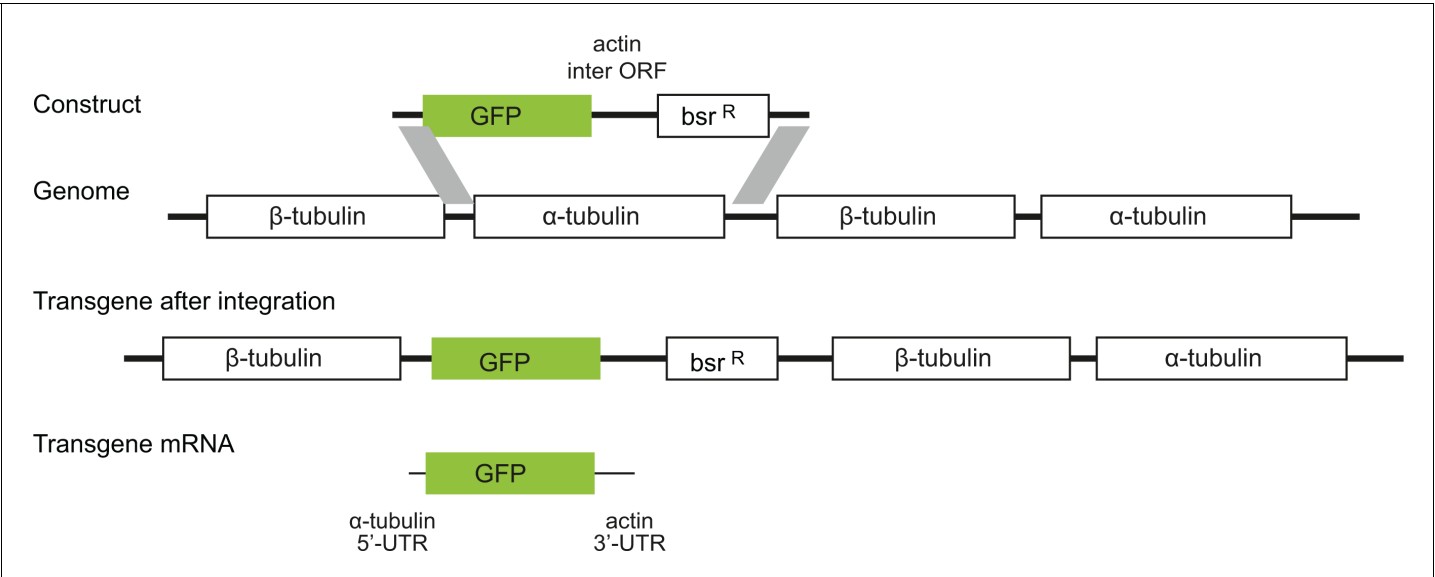

**Figure 2.** Diagram to show the strategy for integration of the GFP transgenes into the tandem repetitive α-tubulin and β-tubulin array.  In every case the transgene mRNA had a α-tubulin 5'UTR and an actin 3'UTR and was expressed by endogenous transcription of the array.

DOI: https://doi.org/10.7554/eLife.32467.005

The following figure supplements are available for figure 2:

**Figure supplement 1.** Differences in expression levels of four GFP transgenes with different codon usage.

DOI: https://doi.org/10.7554/eLife.32467.006

**Figure supplement 2.** Comparison of GFP expression measured as fluorescence by flow cytometry and protein by western blotting.

DOI: https://doi.org/10.7554/eLife.32467.007

measurement of GFP by flow cytometry and estimates of GFP expression from western blotting was $R^2 = 0.965$; validating the use of GFP fluorescence as a measure of total GFP protein.

## Gene expression codon adaptation index, geCAI, a codon usage statistic to predict mRNA levels

In initial experiments using transgenes with different ORFs, the expression of GFP measured by flow cytometry did not correlate as well as might be expected with CAI (*Carbone et al., 2005*) or tAI (*Tuller et al., 2010*) scores (described in more detail below). We decided to re-visit the calculation of the association between codon scores and expression levels. Building on the observation that altering codon use affected both protein and mRNA levels for the MS2bp-GFP-nls transgene described above, the approach taken was to make an assumption that codon use directly affected mRNA levels for most constitutively expressed genes and to derive a codon statistic for genes that best correlated with their measured mRNA abundance levels.

To estimate the abundance of individual mRNAs, an RNA-Seq analysis of poly(A) enriched RNA was performed with three biological replicates of mid-log phase PCF *T. brucei* cells (*Kelly et al., 2017*) (EBI Array Express E-MTAB-3335). The level of each mRNA was then determined and expressed as the mean of the transcripts per million transcripts (TPM) across all three replicates (*Supplementary file 4*).

To develop a set of numerical codon values that could explain mRNA levels, an approach was used in which the codon usage statistic learnt a codon value by maximising the Spearman's rank correlation coefficient between the measured mRNA level of a set of transcripts and the codon usage statistics for the cognate ORFs. Spearman's rank correlation coefficient was used to avoid distributional assumptions about the hidden distribution of per-gene translational efficiencies and the observed distribution of mRNA abundance estimates. To calculate the codon values, the measured expression levels of mRNAs from 5136 single copy genes were used. This set of genes represented the genome with the following exclusions: (i) mRNAs with >2 fold differential expression when BSF and PCF trypanosomes were compared (*Kelly et al., 2017*) as the mRNA abundance are likely to be

modulated by *cis*-acting RNA binding proteins and thus obscure the codon score learning process. (ii) mRNAs encoded by single copy genes without a homologue in at least one other kinetoplastid species, this was a mechanism to exclude any potentially spurious ORFs that may not encode proteins. (iii) mRNAs encoded by multicopy genes families as the accuracy of TPM calculations is compromised by uncertainty of copy number and associated errors in allocation of many sequence reads to individual genes.

The calculation of codon weights $C_i$ proceeded by randomly generating a set of codon scores $S$ were $C_i$ [0,1]. Here, a score of 1 means that a codon has the maximum positive effect on mRNA levels and a score of 0 the minimum effect. The geCAI score of an ORF was evaluated as the geometric mean of the scores for all codons in that ORF. The Spearman's rank correlation coefficient between geCAI scores for the 5136 ORFs and mRNA abundances was then computed. A Markov chain Monte Carlo algorithm was then employed to introduce stochastic changes into the codon score matrix and the entire set of genes was rescored and the Spearman's rank correlation coefficient recalculated. If in a given generation of the Markov chain, a matrix was found that produced a higher correlation coefficient than the current best matrix, then it replaced the current best matrix and formed the basis for subsequent generations of stochastic modification. One thousand chains each starting from a different random starting matrix were initiated and allowed to run for 5000 generations. In all cases, stationary phase was reached between 1500 and 2000 generations (*Figure 3A*). The gene expression codon adaptation index (geCAI) values for each codon were calculated as the median of the values obtained from the 1000 chains (*Figure 3B* and *Table 1*).

When mRNA abundance expressed as TPM was plotted against the ORF geCAI scores (*Supplementary file 4*), the Spearman's rank correlation coefficient for the 5136 mRNAs was $\rho = 0.55$ (*Figure 3C*). A mid-log phase PCF cell contains approximately 50000 mRNA molecules (*Dhalia et al., 2006*) and so 20 TPM is equivalent to one mRNA molecule/cell, so the range of transcript abundance for the vast majority is between ~1 and 40 mRNAs/diploid gene pair/cell. The use of geCAI allows a prediction of a range of expression level for an mRNA based on the sequence of the ORF alone, for example an ORF with a geCAI score of 0.35 will be expressed at 2.5 mRNAs/cell on average with the vast majority of mRNAs in a range between 1 and 5.

Several previously developed codon usage statistics were tested to determine if these could explain the measured mRNA levels. Of the tested methods, CAI (*Carbone et al., 2005*) and tAI (*Tuller et al., 2010*) provided the best correlation but the predictive capacity was limited with Spearman's rank correlation coefficients of $\rho = 0.16$ and $\rho = 0.13$ respectively (*Figure 3—figure supplement 1*). These measures may be unsuitable for this type of analysis as each assumes the effectiveness of the 'best' codon for each amino acid is equivalent, for example that the most efficient codon for glycine will have the same effect as the most efficient codon for alanine. There is evidence that this is not the case (*Gardin et al., 2014*).

The geCAI values for each codon in *Table 1* are derived from the mRNA levels measured in the transcriptome analysis used in this study. There is variation in the measurements of mRNA levels in transcriptome analyses from different studies that probably has a range of origins: identity of the cell line, growth conditions, methodology of RNA preparation, RNA-Seq and data analysis and similar variation has been found in studies in yeast (*Harigaya and Parker, 2016*). The consequence is that any table of geCAI codon values contains an element reflecting how mRNA levels were measured, but the underlying principle that codon choice is a major determinant of mRNA levels is unaffected. To illustrate this point, transcriptome data from three other studies were analysed to determine geCAI values as above. The Pairwise Pearson correlation (r) between $\log_2$(mRNA abundance) measures from this study and three others ranged from 0.68 to 0.86 (*Supplementary file 5A*). The calculation of geCAI values for each codon showed variation derived from the differences in transcriptome measurements (*Supplementary file 5B and C*). Each set of geCAI codon values was then used to derive geCAI scores for each of the 5136 mRNAs and these were plotted against the mRNA expression level determined in the cognate study (*Figure 3—figure supplement 2*). In each case, there was still a relationship between geCAI scores for ORFs and expression levels with a range of Spearman's rank correlation coefficients from $\rho = 0.35$ to $\rho = 0.50$.

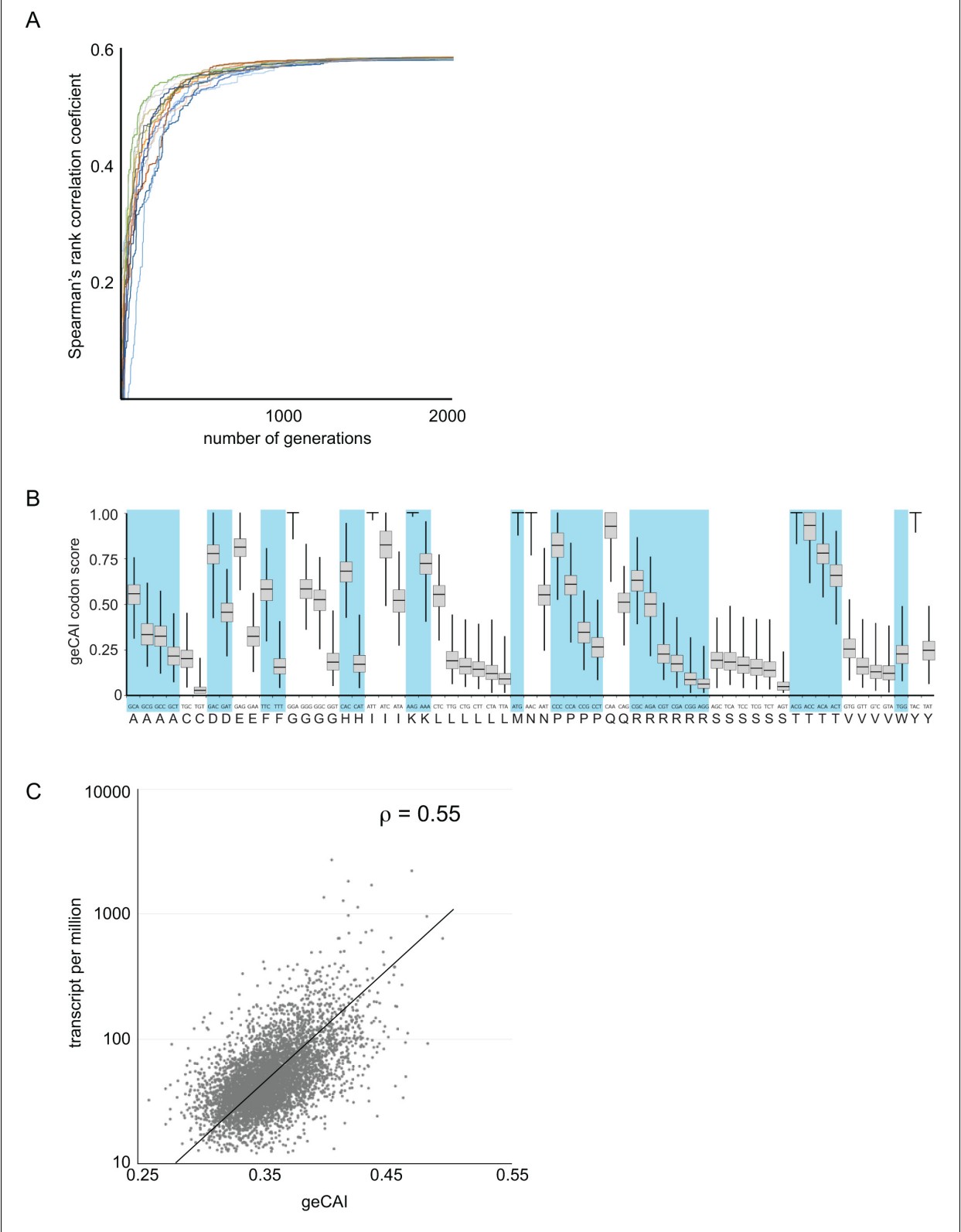

**Figure 3.** Development of a gene expression codon adaptation index (geCAI). Randomly assigned codon values were generated and the Spearman's rank correlation score between the ORF geCAI score (geometric mean of the individual codon values) and the measured cognate mRNA expression levels calculated. Changes were made to the codon values using a Markov chain Monte Carlo algorithm and if the step increased the correlation coefficient it was permitted and the process moved on to the next generation. 5000 generations were used and the process run 1000 times each

*Figure 3 continued on next page*

*Figure 3 continued*

starting with different sets of randomly assigned codon scores. (**A**) The maximisation of the Spearman's rank correlation coefficient between geCAI scores and cognate mRNA expression levels reached a plateau after 1000 to 1500 generations in individual chains, seen here in a plot of Spearman's rank correlation coefficient against generation number for a sample of 20 Markov chain Monte Carlo chains. (**B**) The geCAI values for each codon shown were calculated as the median of the 1000 parallel processes. They are shown here as a box plot with the median and quartiles of each value derived from the 1000 chains. (**C**) Final correlation between geCAI score for each ORF and the measurement of cognate mRNA abundance for the 5136 single copy, non-developmentally regulated genes in the genome. The Spearman's rank correlation coefficient was ρ = 0.55.

DOI: https://doi.org/10.7554/eLife.32467.008

The following figure supplements are available for figure 3:

**Figure supplement 1.** Correlation between (**A**) tAI score and (**B**) CAI score for each ORF and the measurement of cognate mRNA abundance for the 5136 single copy, non-developmentally regulated genes in the genome.

DOI: https://doi.org/10.7554/eLife.32467.009

**Figure supplement 2.** Correlation between mRNA expression levels and geCAI scores calculated from transcriptomes from other studies (*Fadda et al., 2014*; *Christiano et al., 2017*; *Hutchinson et al., 2016*).

DOI: https://doi.org/10.7554/eLife.32467.010

## geCAI score predicts GFP protein and mRNA levels over a range similar to endogenous transcripts

Twenty-two different GFP ORFs (*Supplementary file 3*) were used to systematically test the predictions of expression levels made by geCAI scores. For GFP protein, expression level was measured by flow cytometry of three independent clones for each GFP transgene, in every case there was less than 5% variation in GFP protein abundance between the three clones (*Supplementary file 6A and B*). The GFP protein abundance was expressed relative to eGFP and plotted against geCAI score (*Figure 4A*); the geCAI score was able to predict GFP protein abundance with 84% accuracy (Pearson's correlation coefficient, $r^2 = 0.84$). This compared favourably with the correlation coefficients between GFP protein abundance and either CAI ($r^2 = 0.25$) or tAI ($r^2 = 0.61$) (*Supplementary file 6B*).

Next, the mRNA abundance for four GFP transgenes with different coding sequences with a range of geCAI scores was determined by RNA-Seq analysis. Equal numbers of cells of the four cell lines were mixed, RNA extracted and expression levels for each GFP mRNA measured in transcripts per million transcripts (TPM). Three separate analyses were performed using three independent clones for each GFP transgene. In analysing the results, the measured expression level of each GFP mRNA was adjusted to compensate for the effect of gene copy number and the dilution effect from the mixing of four different cell lines. Trypanosomes are diploid and any endogenous single copy gene will be present in two copies per cell, each of the four cell lines contained a haploid copy of a different GFP transgene so in the mixed population used to prepare RNA there are eight copies of an endogenous gene for one copy of each of the individual GFP transgenes, effectively an 8-fold lower copy number. After adjusting for the effective difference in copy number within the sequenced pool, GFP mRNA abundance was plotted against geCAI values (*Figure 4B*). The Pearson's correlation coefficient between GFP mRNA levels and geCAI was $r^2 = 0.92$ (*Figure 4B* and *Supplementary file 7*). Therefore, as the genomic integration site, UTRs and amino acid sequence were all identical, codon use was the major determinant of steady state levels for the transgene derived GFP mRNA and protein.

The ~25 fold range of GFP mRNA levels, 2.5 to 60 mRNAs/diploid gene pair equivalent/cell (*Figure 4B*) is similar to the range present in mRNAs from single copy genes in the transcriptome. The measurements for the GFP mRNAs are in the higher end of the endogenous transcriptome, probably because the range of GFP geCAI scores included values higher than those present in most endogenous single copy genes. Thus, altering codon use can account for the range of steady state level of most mRNAs and the geCAI score is a good predictor of mRNA level for single copy genes not subject to developmental regulation. Furthermore, discrepancy between geCAI score for a gene and the observed mRNA or protein abundance for that gene is an indicator that other regulatory mechanisms, such as *cis*-acting RNA binding proteins may be acting on that gene to modulate mRNA abundance. mRNAs for cytosolic ribosomal proteins have significantly higher geCAI scores than those encoding mitochondrial ribosomal proteins

**Table 1.** geCAI values for each codon for logarithmically growing procyclic form *T. brucei*.

| Amino acid | Codon | geCAI codon weight | Amino acid | Codon | geCAI codon weight |
|---|---|---|---|---|---|
| A | GCA | 0.55 | N | AAC | 1.00 |
| A | GCG | 0.33 | N | AAT | 0.55 |
| A | GCC | 0.32 | P | CCC | 0.82 |
| A | GCT | 0.21 | P | CCA | 0.61 |
| C | TGC | 0.20 | P | CCG | 0.34 |
| C | TGT | 0.02 | P | CCT | 0.26 |
| D | GAC | 0.78 | Q | CAA | 0.93 |
| D | GAT | 0.45 | Q | CAG | 0.51 |
| E | GAG | 0.81 | R | CGC | 0.63 |
| E | GAA | 0.32 | R | AGA | 0.50 |
| F | TTC | 0.58 | R | CGT | 0.22 |
| F | TTT | 0.15 | R | CGA | 0.17 |
| G | GGA | 1.00 | R | CGG | 0.08 |
| G | GGG | 0.58 | R | AGG | 0.06 |
| G | GGC | 0.52 | S | AGC | 0.19 |
| G | GGT | 0.18 | S | TCA | 0.18 |
| H | CAC | 0.68 | S | TCC | 0.16 |
| H | CAT | 0.17 | S | TCG | 0.15 |
| I | ATT | 1.00 | S | TCT | 0.13 |
| I | ATC | 0.82 | S | AGT | 0.04 |
| I | ATA | 0.52 | T | ACG | 1.00 |
| K | AAG | 1.00 | T | ACC | 0.93 |
| K | AAA | 0.72 | T | ACA | 0.78 |
| L | CTC | 0.55 | T | ACT | 0.66 |
| L | TTG | 0.19 | V | GTG | 0.25 |
| L | CTG | 0.15 | V | GTT | 0.15 |
| L | CTT | 0.14 | V | GTC | 0.13 |
| L | CTA | 0.12 | V | GTA | 0.12 |
| L | TTA | 0.09 | W | TGG | 0.22 |
| M | ATG | 1.00 | Y | TAC | 1.00 |
| | | | Y | TAT | 0.24 |

DOI: https://doi.org/10.7554/eLife.32467.011

The mRNA abundance measurements used to obtain geCAI codon weights did not include mRNAs for 67 of the 75 cytosolic ribosomal proteins as these are encoded by gene families with two or three members. Comparison of the geCAI scores for mRNAs encoding cytoplasmic and mitochondrial ribosomal proteins provided a test of geCAI scores for endogenous genes with orthologous function but different expression levels, there are many more ribosomes in the cytoplasm than in the mitochondrion. Lists of ribosomal protein genes were derived from the structures of both types of ribosome (*Hashem et al., 2013*; *Zíková et al., 2008*) and geCAI scores calculated for each mRNA encoding a ribosomal protein. The more abundant cytosolic ribosomal protein mRNAs had higher geCAI scores than the less abundant mitochondrial ribosome protein mRNAs (*Figure 5* and *Supplementary file 8*). The mean geCAI scores (±SD) were 0.439 (±0.034) for cytoplasmic and 0.378 (±0.033) for mitochondrial ribosomal protein mRNAs respectively. The probability of this difference arising by chance is <0.00001 (unpaired two-sample t test with equal variance). These measurements are consistent with more abundant proteins are encoded by mRNAs with higher geCAI scores.

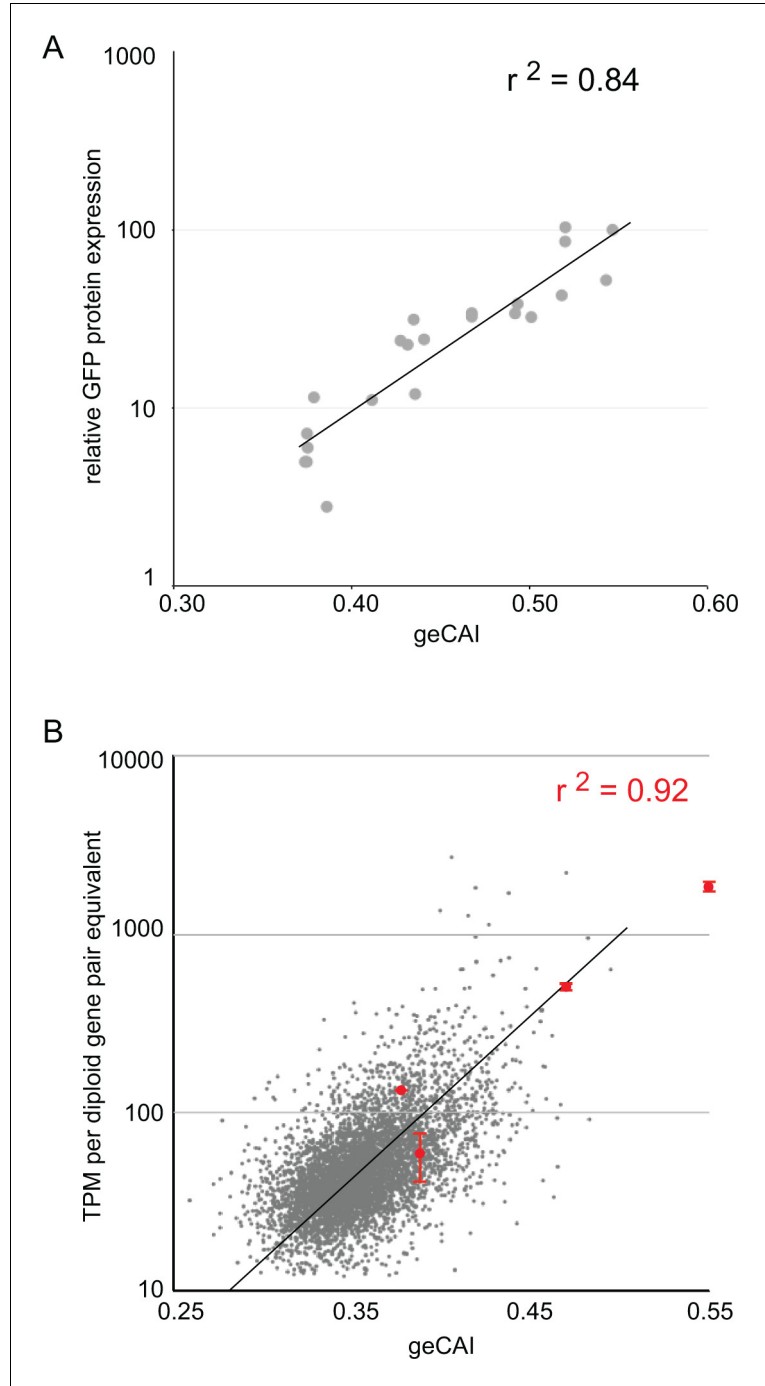

**Figure 4.** geCAI score is a predictor of protein and mRNA expression levels. (**A**) The correlation between $\log_{10}$(GFP protein levels) and geCAI score for twenty-two different ORFs with a range of geCAI scores. The Pearson's correlation coefficient was $r^2 = 0.84$. (**B**) The correlation between geCAI score and $\log_{10}$(GFP mRNA levels) for four different GFP ORFs. The values for the four GFPs are shown in red (±standard error), and are superimposed upon a plot of the endogenous mRNAs in grey, from *Figure 3C*. The Pearson's correlation coefficient was $r^2 = 0.92$.

DOI: https://doi.org/10.7554/eLife.32467.012

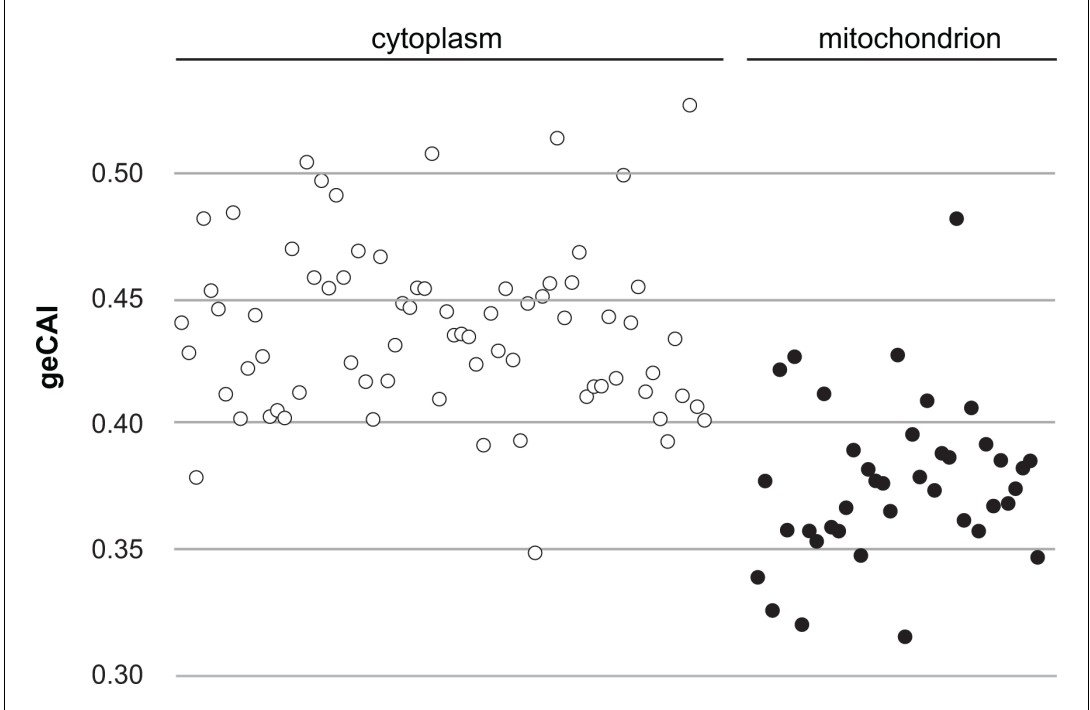

**Figure 5.** A comparison of geCAI scores for mRNAs encoding 75 abundant cytosolic ribosomal proteins and 41 less abundant mitochondrial ribosomal proteins. As 67/75 of the cytosolic ribosomal proteins are encoded by >1 gene, they were excluded from the calculation of geCAI values for individual codons. mRNAs encoding cytosolic ribosomal proteins are shown as open circles and mitochondrial ribosomal proteins as solid circles. The cytosolic ribosome protein ORFs have a significantly higher geCAI score than the mitochondrial ribosome protein ORFs (p<0.00001 unpaired two-sample t test with equal variance).

DOI: https://doi.org/10.7554/eLife.32467.013

## geCAI is a predictor of mRNA turnover rates

The origin of the differences in GFP mRNA levels from transgenes with differently coded ORFs was investigated by measuring turnover of five different GFP mRNAs by quantitative northern blotting after the inhibition of *trans*-splicing, and thus mRNA maturation, with sinefungin (*Pugh et al., 1978*) (*Figure 6A*). The two GFPs with the highest geCAI scores (eGFP: 0.547 and GFP P3: 0.520) had no detectable decay over the time course of 60 min, these both have geCAI scores greater than nearly all endogenous mRNAs. The three other GFPs with lower geCAI scores (GFP P2: 0.493, GFP 226: 0.432 and GFP 102: 0.375) had a rate of decay inversely proportional to the geCAI value (*Figure 6B* and *Supplementary file 9*). For these three GFP mRNAs decay to 50% occurred between 10 and ~80 min, similar to the range of half-lives reported for endogenous mRNAs (*Fadda et al., 2014*).

One study has estimated the half-lives of most mRNAs in procyclic form trypanosomes (*Fadda et al., 2014*), and these measurements were plotted against geCAI scores determined from the cognate transcriptome and this study (*Figure 6—figure supplement 1*). The Spearman's rank correlation coefficient was ρ = 0.22 for the cognate geCAI values and ρ = 0.33 for the geCAI values used in this study. Thus, geCAI scores calculated using either study can explain a significant proportion of variance in mRNA turnover rates at a transcriptome wide level.

## Preventing translation through a hairpin in the 5'UTR blocks geCAI score-mediated instability

The observations above provide evidence that codon use is a major determinant of mRNA half-life, which in turn implies translation is involved. This was tested directly by blocking or reducing translation through the inclusion of secondary structures in the 5'UTR. Five different GFP transgene constructs with a range of geCAI scores were modified by insertion of a 48 base hairpin (24 base pairs)

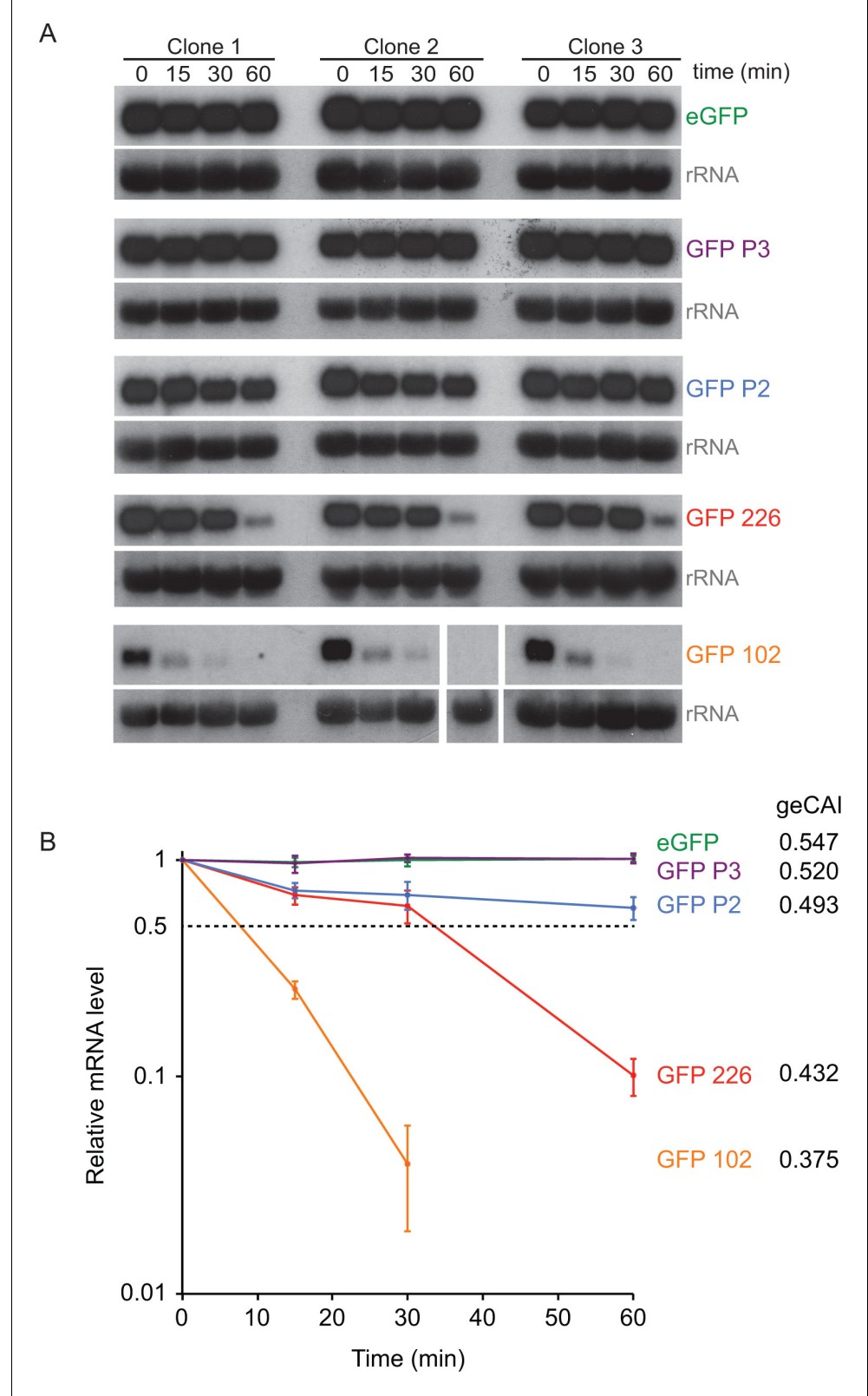

**Figure 6.** geCAI scores are a major determinant of mRNA half-lives. (**A**) Northern blots to measure mRNA levels over a time course after inhibition of mRNA maturation with sinefungin in cell lines expression one of five differently coded GFPs. Three independent clones for each of five cell lines are shown and each blot was probed with the cognate GFP ORF; blots were exposed for a range of times. rRNA was used as a loading control for GFP

*Figure 6 continued on next page*

*Figure 6 continued*

mRNA quantitation. The identity and geCAI score for each GFP ORF is shown to the right. (**B**) Decay of GFP mRNAs after sinefungin addition. The mRNAs were quantified by phosphorimaging of the northern blots and each value adjusted for loading based on rRNA. The mRNA levels (±standard error) were plotted against time after being normalised to the zero timepoint.

DOI: https://doi.org/10.7554/eLife.32467.014

The following figure supplement is available for figure 6:

**Figure supplement 1.** Correlation between mRNA half-lives from (*Fadda et al., 2014*) and geCAI scores for each ORF calculated from the transcriptome data in: (A) reference (*Fadda et al., 2014*); (B) this study.

DOI: https://doi.org/10.7554/eLife.32467.015

in the 5'UTR located 59 nucleotides from the cap and 88 nucleotides from the initiation codon (*Supplementary file 2B*). The same approach has been used previously in trypanosomes to block translation of a specific mRNA (*Webb et al., 2005b*). Measurements of GFP protein and mRNA expression were made in three independent clones for each transgene. As expected, there was no detectable expression of GFP by flow cytometry in any of the cell lines containing a transgene with a hairpin (*Supplementary file 6C and 6D*), therefore the hairpin-containing mRNAs were not translated. Measurements of the mRNA levels by quantitative northern blotting revealed that the effect of inclusion of the hairpin in the 5'UTR was to stabilise the GFP mRNAs (*Figure 7A* and *Supplementary file 6E*) and the lower the geCAI score the greater the relative increase in abundance of the hairpin to control mRNAs from 1.4-fold for eGFP (geCAI = 0.547) to 14-fold for GFP 102 (geCAI = 0.375) (*Figure 7B*). Could this observation be explained by the persistence of decapped mRNA stabilised by the hairpin blocking 5' to 3' decay? This is very unlikely as an mRNA with a similar size and GC-content hairpin in the 5'UTR was readily degraded in procyclic trypanosomes (*Webb et al., 2005b*).

These observations provide evidence that: (i) the translation of an ORF is necessary for the variation in mRNA half-life conferred by the geCAI, (ii) the different GFP mRNA levels result from active destabilisation, and (iii) none of the mRNAs with differently coded GFPs are inherently unstable due to sequence alone.

## Insertion of a shorter secondary structure in the 5'UTR reduces GFP ORF translation but has little effect on state levels of GFP mRNA

The experiments above indicated that translation is necessary for geCAI score-mediated mRNA turnover. The 5'UTR hairpin approach above was modified by decreasing the length of double stranded RNA inserted into the 5'UTR until eGFP protein was expressed. Reduction of the length of the hairpin from 24 bp to 18 bp still blocked translation and no GFP fluorescence was observed. In contrast, a 12 bp stem loop resulted in eGFP protein expression that was 11% of that from the parental transgene without a hairpin in the 5'UTR (*Supplementary file 6F*). The mRNA levels from three independent clones expressing GFP transgenes with the wild type 5'UTR, 18 bp hairpin containing 5'UTR and 12 bp stem loop containing 5'UTR were measured (*Figure 8* and *Supplementary file 6G*), the presence of the 12 bp stem loop resulted in a modest reduction to 76% in the steady state GFP mRNA levels (p=0.13 for a difference between the mRNAs with and without a 12 bp stem loop; unpaired two-sample t test). This observation that a nine-fold reduction in the frequency of translation has only a small effect on the GFP mRNA levels favours a model in which geCAI score is interpreted by the translating ribosome by a mechanism mostly independent of ribosome density/ frequency of translation, possibly the rate of ribosome progression.

Two sets of ribosomal profiling data are available that contain values for ribosome density for >5000 of the 5136 non-developmentally regulated mRNAs expressed from single copy genes that were used to derive geCAI values (*Vasquez et al., 2014*; *Jensen et al., 2014*). There is a correlation between ribosome density and mRNA levels but there is also great variation in the ribosome density on different mRNAs with similar expression levels (*Jensen et al., 2014*) indicating frequency of translation/ribosome density alone is not sufficient to determine translationally regulated mRNA turnover. The geCAI scores of mRNAs were plotted against ribosomes density for the two ribosome profiling datasets (*Supplementary file 10*). The Pearson's correlation coefficient between ribosome footprint levels and geCAI was $R^2 = 0.141$ and $0.244$ for the two sets of ribosome density values.

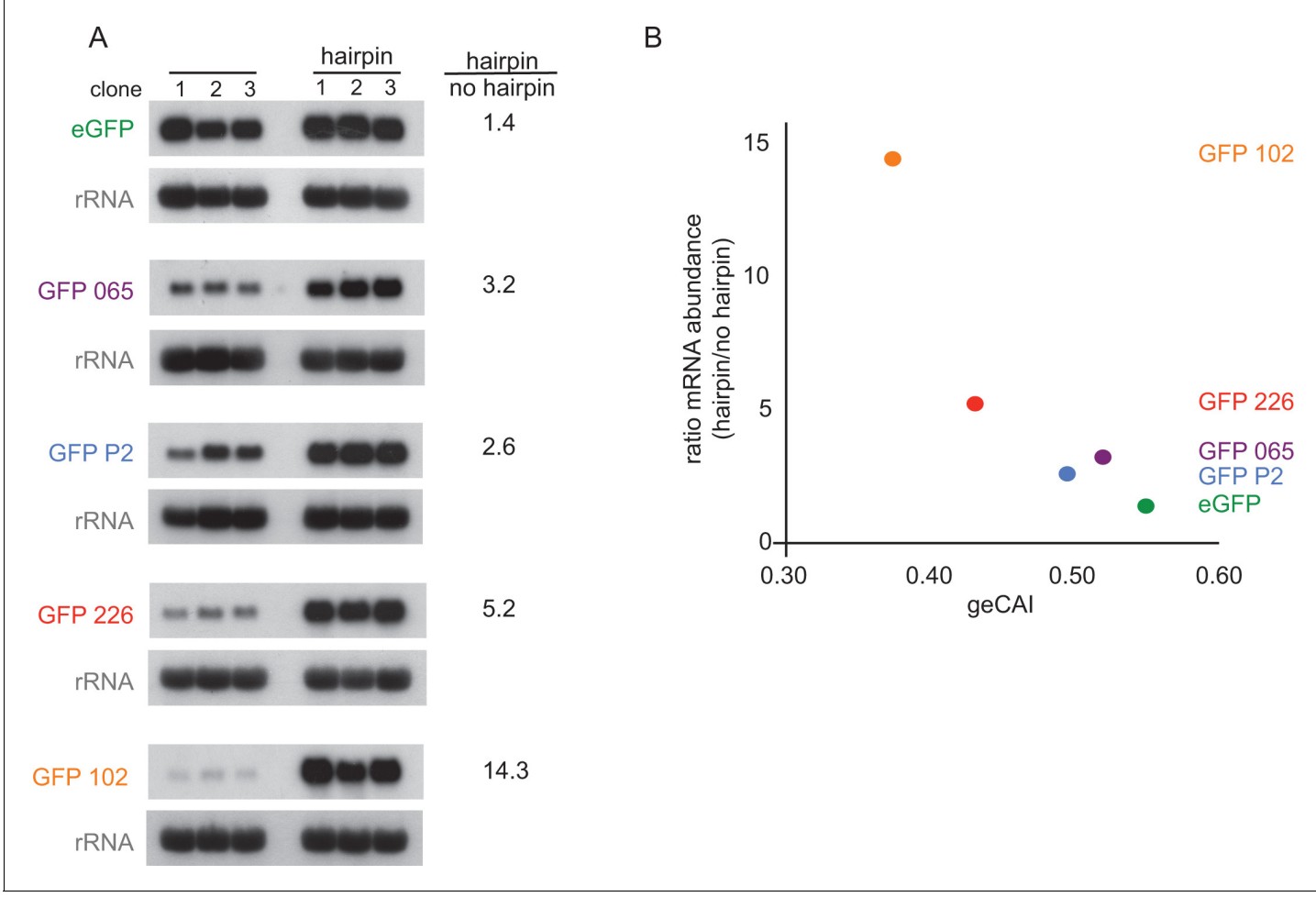

**Figure 7.** Blocking translation stabilises mRNAs with low geCAI scores more than mRNAs with high geCAI scores. (**A**) Northern blots to measure GFP mRNAs in cell lines expressing GFP with geCAI scores with or without a hairpin structure in the 5′UTR that blocked translation. Three independent clones for each transgene were analysed and GFP; mRNA was quantified by phosphorimaging and values adjusted for loading using rRNA. (**B**) Graph showing the fold increase in mRNA steady state level due to the inclusion of the hairpin in the 5′UTR plotted against the geCAI score for each ORF.
DOI: https://doi.org/10.7554/eLife.32467.016

Thus occupancy/frequency of translation can explain a significant component of the geCAI measure, but suggests a model in which geCAI mediated mRNA turnover is also dependent on other factors.

## Many developmentally regulated mRNAs do not conform to geCAI predictions

The geCAI values for each codon were calculated using a set of values for mRNA expression levels that excluded developmentally regulated mRNAs, defined in the case as having a twofold or greater difference in abundance when PCF cells were compared with BSF cells. Analysing only single copy genes, the geCAI scores for 372 mRNAs upregulated in PCF cells, and 176 mRNAs upregulated in BSF cells were calculated and plotted against expression levels in PCF cells (***Figure 9***). mRNAs upregulated in PCFs are largely expressed at higher levels than would be predicted by geCAI alone. In contrast, mRNAs downregulated in PCFs (upregulated in BSFs) are expressed at levels lower than predicted by geCAI. Taken together, these measurements provide evidence that developmental regulation results from both stabilisation and destabilisation pathways that act in concert to modulate the geCAI determined mRNA levels.

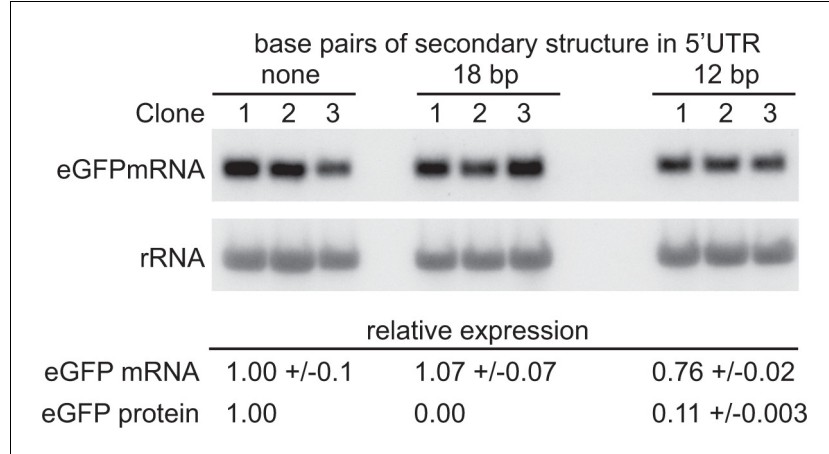

**Figure 8.** Partially blocking translation of a GFP mRNA decreases protein but has only a small effect on mRNA levels. (A) Northern blot to measure GFP mRNAs in cell lines expressing GFP with different secondary structures in the 5'UTR. Three independent clones for each transgene were analysed; mRNA was quantified by phosphorimaging and values adjusted for loading using rRNA. (B) Quantitation of the effect of secondary structures on GFP protein and mRNA levels. GFP protein was measured by flow cytometry (*Supplementary file 6F*) and GFP mRNA by phosphorimaging, rRNA was used to adjust for loading (*Supplementary file 6G*). Values are normalised to the 5'UTR with no added secondary structure. In each case the value is the average of three independent clones and the standard error is shown.
DOI: https://doi.org/10.7554/eLife.32467.017

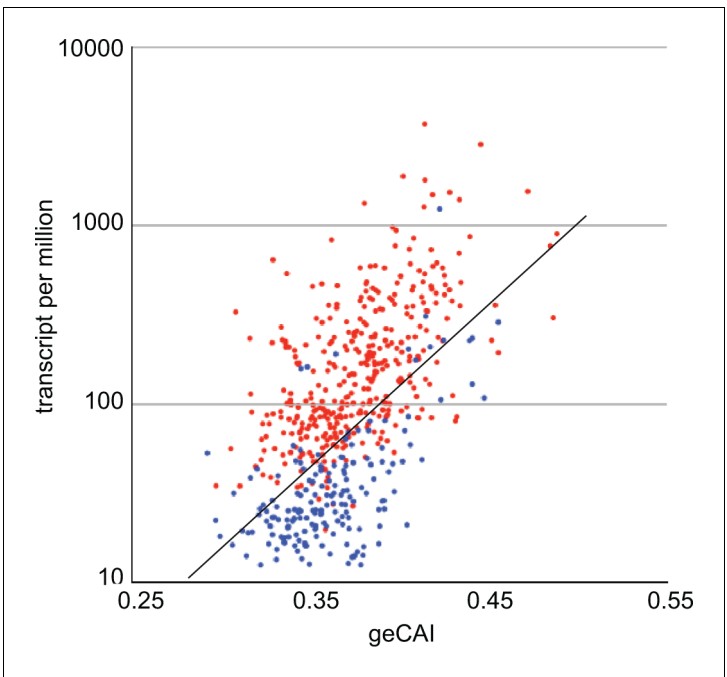

**Figure 9.** geCAI and expression levels of developmentally regulated mRNAs in procyclic forms. Expression levels in procyclic forms plotted against geCAI scores for developmentally regulated mRNAs transcribed from single copy genes. Red: 374 mRNAs more than twofold up regulated in procyclic forms compared to bloodstream forms. Blue: 182 mRNAs more than twofold down regulated in procyclic forms when compared to bloodstream forms. The diagonal line is the best fit line for the 5136 non-developmentally regulated mRNAs (*Figure 3C*).
DOI: https://doi.org/10.7554/eLife.32467.018

## Discussion

Trypanosomatids have evolved separately from other eukaryotic lineages for ~500 million years (**Lukeš et al., 2014**). Over this time the core gene expression machinery present in the last eukaryotic common ancestor has been largely conserved but selective transcription of individual genes by RNA polymerase II has been lost and individual mRNAs arise by co-transcriptional processing of polycistronically transcribed precursors. Tens or hundreds of genes, mostly with no obvious functional link to each other, are transcribed from a single start site and monocistronic mRNAs arise by co-transcriptional *trans*-splicing and polyadenylation. The downstream consequence is that individual mRNA levels must be set post-transcriptionally.

An observation that altering the codon use of a transgene to codons under-represented in highly expressed mRNAs not only reduced protein expression but also the cognate mRNA to a similar extent led us to investigate the effect of codon use on mRNA levels. The major finding presented here is that codon use is central to the regulation of mRNA levels in trypanosomes. A novel codon usage metric, the gene expression codon adaptation index (geCAI), was developed that maximises the relationship between codon usage and the measured mRNA abundance. The geCAI score for an ORF allows a prediction of mRNA levels for constitutively expressed genes. Differently coded GFPs were used as a test and mRNA expression levels were successfully predicted over a ~ 25 fold range, similar to the range of endogenous mRNAs. The differences in GFP mRNA levels resulted from differences in the half-lives of the mRNAs, the lower the geCAI score, the shorter the half-life. An investigation of the mechanism of geCAI mediated mRNA half-life showed that translation was necessary as mRNA turnover was reduced/prevented if translation was blocked using a hairpin in the 5'UTR. Further, insertion into the 5'UTR of secondary structure that greatly reduced but did not block translation had only a small effect on steady state mRNA levels, providing evidence that the sensor for geCAI mediated turnover was detecting ribosome progression rather than frequency of translation initiation.

## Development of geCAI

Initial attempts at a numerical analysis of codon choice and mRNA levels using existing codon metrics were not particularly successful especially for the expression of reporter genes containing codons found infrequently in abundant mRNAs. For example, the most commonly used codon metric is the codon adaptation index (CAI) (**Sharp et al., 1987**) and previous work had found only a weak correlation between CAI and protein levels in both *T. brucei* (**Horn, 2008**) and the related species *Leishmania mexicana* and *L. major* (**Subramanian and Sarkar, 2015**). However, this association between codon use and mRNA abundance suggested that perhaps an alternative metric may have increased explanatory power. The aim in developing geCAI was to account for the full range of mRNA levels present in the procyclic form of *T. brucei*. The measured mRNA levels for the set of single copy, non-developmentally regulated genes were used to compute a Spearman's rank correlation coefficient between transcript abundance and the geCAI score. The final Spearman's rank correlation coefficient between geCAI score and mRNA levels was $\rho = 0.55$, indicating that codon usage is responsible for more than 50% of the variation in the expression of mRNAs from different genes. The main difference between the geCAI and other CAI statistics is that the codon weight matrix is global rather than local. Specifically, for a conventional CAI matrix one codon for each amino acid must be assigned a value of 1, other synonymous codons for each amino acid must have values $\leq 1$ and>0. This makes the assumption that the translation efficiency of the optimal synonymous codon for each amino acid is equivalent. This local constraint (one codon for each amino acid has a value = 1) is not enforced when calculating geCAI but a single global constraint that stipulates that at least one from the 61 codons must have the maximum value of 1 and that all other codons must have values $\leq 1$ and>0. Thus, unlike other codon weight matrices, such as CAI (**Sharp et al., 1987**) and tAI (**dos Reis et al., 2004**), the optimal synonymous codon for one amino acid does not necessarily have the same value as the optimal synonymous codon for any other amino acid. This more accurately represents the real relationship between codons and thus enables greater explanatory power.

In the set of geCAI values for all amino acids, only six codons have a value of 1 and there is also an unexpected variation in values between codons encoding similar amino acids. For example, all six codons for serine have geCAI values below 0.25 whereas threonine codons range from 0.75 to 1

(*Figure 3B*). The selection for these contrasting geCAI values for similar amino acids is not obvious but may be related to maintaining most mRNAs at low copy number.

## geCAI and mRNA abundance

The geCAI scores provided a prediction of a range mRNA abundance, for example the vast majority of mRNAs encoding ORFs with a geCAI score of 0.35 vary between 1 and 5 mRNA molecules per cell. This range of mRNA levels suggests that there are determinants of the final mRNA levels that lie outside codon use: first, through the action of *cis*-elements and *trans*-acting factors the alter mRNA stability directly. Second, it is not known whether all the polycistronic transcription units are transcribed at the same rate, so some of the variation might arise from differential transcription rates. Third, the cohort of single copy genes will include some that are developmentally regulated in one of the several other developmental forms not considered here. However, the predictive ability of geCAI scores for mRNA expression level is remarkable and, along with the experiments altering translation of the reporter mRNA, provides good evidence that the half-life of most mRNAs is set by a translation associated process.

Developmentally regulated mRNAs provide an example of cohorts of mRNAs that alter in response to external signals. In the PCFs used in experiments here, the expression levels of the sets of developmentally regulated mRNAs do not correlate well with geCAI scores. Many mRNAs down regulated in PCFs (when compared to BSFs) were expressed at levels lower than predicted by geCAI scores. This might be expected if the intrinsic geCAI expression level of these mRNAs is modulated by *cis*-elements in the UTRs to destabilise the mRNA (*Webb et al., 2005a*; *Clayton, 2013*). Conversely, many mRNAs upregulated in PCFs relative to BSFs were expressed at levels greater that would be predicted by the geCAI score. This indicates that these mRNAs are actively stabilised in PCFs and that developmental regulation comes from a combination of active stabilisation and destabilisation that modulates geCAI score mediated stability. Two comparative analyses of BSFs and PCFs by ribosome profiling showed that regulated translation was common (*Vasquez et al., 2014*; *Jensen et al., 2014*). Thus, could inhibition or stimulation of translation directly cause alteration to mRNA levels? The evidence here where reducing the frequency of translation had little effect on mRNA levels indicates the two are not necessarily linked. Further, the weak correlation between geCAI score for an ORF and the ribosome density indicates that abundant mRNAs (high geCAI score) are not necessarily translated frequently. However, should a *trans*-acting factor act in a similar manner to Fragile X mental retardation protein (*Darnell et al., 2011*) and cause ribosome slowing or stalling during translation then an effect on mRNA stability could follow.

The differential speed of progression of a ribosome on different codons is generally held to be dependent on the supply of cognate charged tRNAs (*Fluitt et al., 2007*; *Chu et al., 2011*; *Chu et al., 2014*) and this forms the basis of the tRNA adaptation index (tAI) (*dos Reis et al., 2004*). The availability of charged tRNAs in vivo is not easily determined and approximations using the copy number of genes for each tRNA (the larger the copy number, the more tRNA) do not take account of post-transcriptional modifications to the tRNA that alter codon binding (*Novoa and Ribas de Pouplana, 2012*); the latter has been used to improve the interpretation of codon bias (*Novoa et al., 2012*). In yeast there are >270 tRNA genes, whilst in the *T. brucei* genome there are only 66 tRNA genes including two for selenocysteine (*Tan et al., 2002*; *Padilla-Mejía et al., 2009*). The partially characterised modifications of tRNAs in trypanosomes (for example [*Rubio et al., 2007*; *Rubio et al., 2006*; *Gaston et al., 2007*; *Bruske et al., 2009*; *Ragone et al., 2011*; *Krog et al., 2011*]) and the uncertain effect of base modifications and wobble on efficiency of decoding remains unclear and means it is not trivial to relate the geCAI value for each codon to the cognate tRNA abundance.

## Comparison with yeast and animals

Codon optimality has also been shown to be a major determinant of mRNA stability in yeast ([*Presnyak et al., 2015*] and reviewed in [*Heck and Wilusz, 2018*]) and operates through the detection of slower ribosome progression by the RNA helicase DHH1 (*Radhakrishnan et al., 2016*) which in turn can activate mRNA decay (*Coller et al., 2001*; *Sweet et al., 2012*). The findings here provide evidence that a similar process that links speed of ribosome progression to mRNA decay operates in trypanosomes. An orthologue of DHH1 is present in trypanosomes (*Schwede et al., 2008*;

*Kramer et al., 2010*) and co-precipitates with the NOT deadenylase complex (*Schwede et al., 2008*; *Färber et al., 2013*). Moderate over expression of DHH1 resulted in altered levels of developmentally regulated mRNAs and a slowing of growth (*Kramer et al., 2010*) but little else is known and unlike yeast, DHH1 is essential in trypanosomes.

In several animals, maternal mRNAs are marked by codon use for degradation at the maternal-to-zygotic transition and this may reflect changes in tRNA availability or re-activation of the degradation pathway (*Mishima et al., 2016*; *Bazzini et al., 2016*). Changes in tRNA pools in response to external stimuli is an attractive model for regulating gene expression in developmental transitions and deletion of individual tRNA genes can alter mRNA levels in yeast (*Bloom-Ackermann et al., 2014*). Direct measurements of the tRNA pools in trypanosomes would inform whether a similar change underpins any of developmental transitions in trypanosomes.

In summary, we have developed and validated a new codon use metric, the gene expression codon adaptation index (geCAI) and shown that it can be used to predict the expression level of many mRNAs in PCF trypanosomes. The lack of selective regulation of transcription of most protein coding genes in trypanosomes means that the effect of codon use on mRNA levels is particularly apparent. Mechanistically, the evidence supports a model in which the speed or processivity of ribosome progression determines mRNA half-life. This is similar to the process identified in yeast and, if there is a common mechanism in yeast and trypanosomes, it is likely that it has a very early evolutionary origin possibly dating from the last eukaryotic common ancestor, and that it may be present in many diverse eukaryotes.

## Materials and methods

### Gene expression codon adaptation index

A script to enable estimation of geCAI codon weights is available for download under the GPL V3.0 licence at https://github.com/SteveKellyLab/geCAI (*Kelly, 2017*; copy archived at https://github.com/elifesciences-publications/geCAI).

A codon weight matrix was generated with each codon randomly assigned a weight in the interval [0, 1] with a constraint that at least one codon had a value of 1. These codon values were used to calculate a score each ORF as the geometric mean of the constituent codon values. The Spearman's rank correlation coefficient between the transcript abundance estimates and the ORF scores was then computed. The Spearman's rank correlation coefficient was then maximised by using a Markov chain Monte Carlo algorithm to introduce stochastic changes into the codon weight matrix and selecting matrices that led to increased Spearman's rank correlation coefficients. 1000 chains were run for 5000 generations and the final codon geCAI value was calculated as the median value of the 1000 chains.

### RNAseq for reference transcript abundances

Three biological replicates of *T. brucei* TREU 927 procyclic form cells were grown in SDM-79 (*Brun and Jenni, 1977*) and harvested at $6 \times 10^6$/ ml and RNA prepared using the Qiagen RNAeasy kit. Cells were washed once with ice cold PBS before the lysis step in RNA preparation procedure. This lysis step was performed within 5 min of removal from the growth incubator. The cDNA libraries were prepared and sequenced at the Beijing Genomics Institute (Shenzhen, China) (*Fiebig et al., 2015*). In brief, polyadenylated RNA was purified from total RNA, converted to cDNA using random hexamer primers sheared and size selected for fragments ~ 200 bp in length using the Illumina Tru-Seq RNA Sample Preparation Kit v2. RNAseq of the resulting libraries was used for the determination of transcript abundances. Sequencing was performed on an Illumina Hiseq 2000 (Illumina, CA) platform. Paired end reads were subject to quality trimming and adaptor filtering using Trimmomatic (*Bolger et al., 2014*) using the settings 'LEADING:10 TRAILING:10 SLIDINGWINDOW:5:15 MIN-LEN:50'. Trimmed reads were then quantified against the *T. brucei* TREU927 v6 transcripts using RSEM (*Li and Dewey, 2011*) using the default settings for RSEM. Bowtie 2 (*Langmead and Salzberg, 2012*) is used as part of the RSEM protocol using setting described in the paper (*Li and Dewey, 2011*). For computing of geCAI scores all transcript abundance estimates (Transcripts per million transcripts) were averaged across the three biological replicates. The transcript reads are in

EBI ArrayExpress E-MTAB-3335 and developmental regulation of mRNA expression was derived from a comparison of data in EBI ArrayExpress E-MTAB-3335.

## GFP ORF sequences

eGFP was purchased from Clontech (Takara, Kyoto, Japan). GFPs 71, 183, 188, 194, 226, 102, 163, 205, 211 and 224 were kindly provided by Grzegorz Kudla (*Kudla et al., 2009*). All others GFP sequences were commercially synthetized (Eurofins MWG Operon). For all GFP sequences used see *Supplementary file 3*.

## Plasmids and cloning

All plasmids and oligonucleotides used in this work are described in *Supplementary file 11*. All GFP sequences were cloned into the vector p3827 using HindIII and BamHI restriction sites. Plasmid p3827 is designed to integrate transgenes into the *T. brucei* tubulin locus which contains a long tandem array of alternating alpha and beta tubulin genes (*Figure 2*). Digestion with PacI released a fragment that has the following components in order: alpha to beta tubulin inter-ORF sequence, transgene ORF; actin inter-ORF sequence, blasticidin resistance ORF, alpha to beta tubulin inter-ORF. After integration, the transgene is transcribed from the distant endogenous RNA polymerase II promoter. In order to generate a hairpin in the 5'-UTR of a GFP transgene, p4432 (*Supplementary file 11*) was first modified by the addition of an EcoRV restriction site using BglII and HindIII sites and phosphorylated oligonucleotides D799, D780, D781 and D782, making the plasmid p4699. After, p4699 was linearized with EcoRV and ligated with the phosphorylated oligonucleotides D797 and D798, generating p4724. GFP ORFs for GFP 102, GFP 226, GFP P2 and GFP 065 were cloned into p4724 using HindIII and BamHI restriction sites. Similarly, in order to generate shorter hairpins on the 5'-UTR of the eGFP, p4699 was linearized with EcoRV and ligated with the phosphorylated oligonucleotides E417 and E418 generating p4841, or E419 and E420, generating p4842. Standard procedures were used for generating all plasmids, the sequences of the GFP transcript, modifications to the 5'UTR and p4432 are in *Supplementary file 2*.

## Cell culture and transfections

All experiments were performed using the procyclic developmental form of *Trypanosoma brucei* Lister 427 KG (a kind gift of Keith Gull). Cells were cultured in SDM-79 at 27°C and 5% CO$_2$. Electroporations were as described in (*McCulloch et al., 2004*) using 5–10 µg of PacI digested plasmid. 10 µg/ml of blasticidin was use to select the transfectants. Independent clones were picked on the tenth day after transfection.

## Flow cytometry

Three independent clones of each cell line, growing in mid-logarithmic phase, were analysed by flow cytometry. Data was acquired using FACScan (Becton Dickinson, Franklin Lakes, NJ) and analysed using Cell Quest V3.3 software. Flow Check Fluorospheres (Beckman Coulter, Pasadena, CA) were used to normalise the readings from experiments performed on different days.

## Western blots

Protein samples were loaded in 17.5% SDS-PAGE and transferred to Immobilon membrane (Millipore, Burlington, MA). GFP Rabbit IgG Polyclonal Antibody Fraction (Life Technologies, Waltham, MA) was used to detect eGFP and a cross-reacting band was used as loading control. Second incubation was with peroxidase conjugated donkey anti-rabbit IgG. Detection was carried by enhanced chemiluminescence (ECL) using Fuji Medical X-Ray Film. Films were digitalised and the signal was quantified using ImageJ 1.48 v.

## RNA sequencing and northern blots

RNA was prepared from cultures with cell densities between 4 and 6 $\times$ 10$^6$/ ml using the RNeasy Mini Kit (Qiagen, Hilden, Germany) and quantified using Nanodrop and ethidium bromide stained agarose gels. All experiment used three biological replicates with independent clones for each cell line. Cells were washed once with room temperature serum-free culture medium before the lysis step in RNA preparation procedure. This lysis step was performed within 4 min of removal from the

growth incubator. For the measurement of GFP mRNA levels, equal numbers of cells expressing eGFP, GFP 226, GFP 102 and GFP S5 were mixed and RNA prepared as above. RNAseq and quantitation was as above. For the quantification of the half-lives of different GFP, cells expressing eGFP, GFP P2, GFP P3, GFP 226, GFP 102 were treated with 2 µg/ml of sinefungin and RNA prepared as above at 0, 15, 30 and 60 min. For northern blots, 3 µg of each sample was loaded in 1.2% agarose gel after denaturation with glyoxal (*Carrington et al., 1987*). Quantitative analyses were done using phosphoimager and ImageQuant TL 1D v7.0 (GE Healthcare). All probes used were the complete ORFs: each GFP was probed specifically with the cognate ORF. 18S ribosomal RNA was used as loading control.

## Acknowledgements

We would like to thank Christine Clayton for discussions, Nancy Standart for a critical reading of the manuscript and Grzegorz Kudla for the gift of a range of GFP ORF clones.

## Additional information

### Funding

| Funder | Grant reference number | Author |
|---|---|---|
| Coordenação de Aperfeiçoamento de Pessoal de Nível Superior | | Janaina de Freitas Nascimento |
| Cambridge Overseas Trust | | Janaina de Freitas Nascimento |
| Biotechnology and Biological Sciences Research Council | | Steven Kelly |
| Horizon 2020 Framework Programme | 637765 | Steven Kelly |
| Wellcome | 085956/Z/08/Z | Mark Carrington |

The funders had no role in study design, data collection and interpretation, or the decision to submit the work for publication.

### Author contributions

Janaina de Freitas Nascimento, Conceptualization, Formal analysis, Funding acquisition, Investigation, Methodology, Writing—original draft; Steven Kelly, Conceptualization, Software, Formal analysis, Funding acquisition, Investigation, Methodology, Writing—review and editing; Jack Sunter, Conceptualization, Investigation, Writing—review and editing; Mark Carrington, Conceptualization, Formal analysis, Supervision, Funding acquisition, Investigation, Methodology, Project administration, Writing—review and editing

### Author ORCIDs

Janaina de Freitas Nascimento (iD) http://orcid.org/0000-0002-1757-3441
Steven Kelly (iD) http://orcid.org/0000-0001-8583-5362
Jack Sunter (iD) http://orcid.org/0000-0002-2836-9622
Mark Carrington (iD) http://orcid.org/0000-0002-6435-7266

### Decision letter and Author response

Decision letter https://doi.org/10.7554/eLife.32467.034
Author response https://doi.org/10.7554/eLife.32467.035

# Additional files

## Supplementary files

• Supplementary file 1. Sequences of the various MS2bp-GFP-nls transgenes used is the experiment in *Figure 1*. Each sequence is split into three: the 5' sequences, the open reading frame in bold, and 3' sequences.

DOI: https://doi.org/10.7554/eLife.32467.019

• Supplementary file 2. GFP transgene mRNA sequences (A) GFP transgene sequence after integration. 5' region from an alpha-tubulin gene, *trans*-splice acceptor sites in red (*Kolev et al., 2010*), GFP open reading frame in green, and 3' region from an actin gene, polyadenylation sites in blue (*Kolev et al., 2010*). Grey sequences added to construct for cloning of GFP variants (B) 5'UTR variants. Sequences adding secondary structure to the 5'UTR are shown in blue; the two characterised trans-splicing site AG acceptor sites are shown in red; the initiation codon is in green and other additional sequences are in grey.

DOI: https://doi.org/10.7554/eLife.32467.020

• Supplementary file 3. Sequences of the 22 GFP open reading frames used in this study, the corresponding plasmids are listed in *Supplementary file 11*.

DOI: https://doi.org/10.7554/eLife.32467.021

• Supplementary file 4. The expression level of each mRNA encoded by a single copy gene in transcripts per million transcripts (TPM). The values for three biological replicates for two developmental forms, PCF procyclic forms and BSF bloodstream forms, are shown. 'P-adj' is the Benjamini-Hochberg adjusted P-value for the BSF measurements being the same as the PCF measurement. The data was taken from EBI Array Express E-MTAB-3335 (*Kelly et al., 2017*).

DOI: https://doi.org/10.7554/eLife.32467.022

• Supplementary file 5. (A) Pairwise Pearson correlation (r) between $\log_2$(mRNA abundance) for single copy genes determined in this and three other studies: A, from reference (*Fadda et al., 2014*); B, from reference (*Christiano et al., 2017*); C from reference (*Hutchinson et al., 2016*). (B) geCAI values for individual codons derived from mRNA abundances from this and three other studies: A, from reference (*Fadda et al., 2014*); B, from reference (*Christiano et al., 2017*); C from reference (*Hutchinson et al., 2016*). (C) Pairwise Pearson correlation (r) between calculated geCAI values calculated from mRNA abundance measurements in this and three other studies: A, from reference (*Fadda et al., 2014*); B, from reference (*Christiano et al., 2017*); C from reference (*Hutchinson et al., 2016*).

DOI: https://doi.org/10.7554/eLife.32467.023

• Supplementary file 6. (A) Table showing flow cytometry measurements of GFP expression in cell line expressing each of the 22 differently encoded GFPs. (B) Summary of GFP expression levels measured by flow cytometry and calculation of correlation coefficients between expression and geCAI, CAI and tAI. (C) Table showing flow cytometry measurements for cell lines containing GFP transgenes with a 24 bp hairpin in the 5'UTR. (D) Summary of GFP expression levels measured by flow cytometry from cell lines containing a GFP transgene with a hairpin in the 5'UTR. (E) mRNA measurements from *Figure 7*. (F) Table showing flow cytometry measurements for cell lines containing GFP transgenes with a a range of secondary structures in the 5'UTR. G. mRNA measurements from *Figure 8*. The rRNA values were used to adjust for loading.

DOI: https://doi.org/10.7554/eLife.32467.024

• Supplementary file 7. Summary of GFP mRNA expression levels measured by RNA Seq and calculation of correlation coefficient between gCAI and mRNA levels.

DOI: https://doi.org/10.7554/eLife.32467.025

• Supplementary file 8. Identification of ribosome proteins from the cytoplasmic (PDB 4V8M) (*Hashem et al., 2013*) and mitochondrial ribosomes (*Zíková et al., 2008*) along with the geCAI scores for each mRNA. The cytosolic ribosome proteins encoded by > 1 gene were similar and the geCAI values is for the first gene in the list.

DOI: https://doi.org/10.7554/eLife.32467.026

• Supplementary file 9. Quantitation of GFP mRNA decay after inhibition of mRNA maturation by sinefungin by phosphorimager analysis of the northern blots in *Figure 6*. The rRNA data was used to adjust for loading.
DOI: https://doi.org/10.7554/eLife.32467.027

• Supplementary file 10. Calculation of the correlation coefficients between geCAI and $\log_{10}$(ribosome footprint measure) for data from two studies. Of the 5136 single copy genes used in this study both the footprinting dataset contained measures for > 5000. Study A is from reference (*Vasquez et al., 2014*) and study B is from reference (*Jensen et al., 2014*).
DOI: https://doi.org/10.7554/eLife.32467.028

• Supplementary file 11. A list of: A, plasmids used in this study and B, oligonucleotides used.
DOI: https://doi.org/10.7554/eLife.32467.029

• Transparent reporting form
DOI: https://doi.org/10.7554/eLife.32467.030

## Major datasets

The following previously published dataset was used:

| Author(s) | Year | Dataset title | Dataset URL | Database, license, and accessibility information |
|---|---|---|---|---|
| Kelly S, Ivens A, Mott GA, O'Neill E, Emms D, Macleod O, Voorheis P, Tyler K, Clark M, Matthews J, Matthews K, Carrington M | 2017 | Continuous culture of bloodstream form and procyclic form of Trypanosoma brucei | https://www.ebi.ac.uk/arrayexpress/experiments/E-MTAB-3335/ | Publicly available at ArrayExpress (accession no. E-MTAB-3335) |

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
