## [Decision Letter]

Thank you for sending your article entitled "Codon choice dictates constitutive mRNA levels in trypanosomes" for peer review at *eLife*. Your article is being evaluated by three peer reviewers, one of whom is a member of our Board of Reviewing Editors and the evaluation is being overseen by a Reviewing Editor and Wendy Garrett as the Senior Editor.

Given the list of essential revisions, including new experiments, the editors and reviewers invite you to respond within the next two weeks with an action plan and timetable for the completion of the additional work. We plan to share your responses with the reviewers and then issue a binding recommendation.

Summary:

This paper shows that in trypanosomes, as in yeast and metazoa, codon usage affects mRNA stability. The authors do this by measuring the abundance and stabilities of variously-coded GFP mRNAs in procyclic trypanosomes. They then use the information, combined with an RNASeq dataset, to calculate optimised scores for the effects of different codons. Importantly, the authors show that translation is required for the codon effects, however they do not demonstrate that the hairpins are really working to block translation as anticipated.

Essential revisions:

1) The authors need to improve the clarity of the paper in every section. Concepts need explaining, and the figures need much more detail in their accompanying legends to be comprehensible.

2) In regard to Figure 1 and Figure 7 as well as the others, the authors must Strengthen/weaken the significance of their analysis by showing that their index works on different RNAseq datasets:

The authors have optimized their CAI for one dataset (developmentally un-regulated genes and tested for mRNA from PF cells). The authors state that their dataset differs from all the others and therefore comparisons are meaningless.

However, the reviewers disagree. If their CAI only works with their own dataset then it is the CAI that is problematic. Also, although they claim that the CAI reflects mRNA half-life, the authors ignore the published transcriptome-wide half-life measurements.

Please show the following scatter plots (essential):a) CAI vs other procyclic form RNASeq datasets. (In addition to the datasets mentioned, there are also some procyclic ones in 10.1093/nar/gkv731, and the original Siegel et al. set.)

b) CAI vs a few bloodstream form datasets (excluding developmentally regulated mRNAs, of course)

c) CAI versus mRNA half-life (Fadda et al., 2014).

The results might be less compelling than those obtained using the dataset with which the CAI was trained, but probably not completely random. Nevertheless, the authors can then discuss the strengths and limitations of their CAI in a balanced way- and also, of course, the limitations of some of the datasets (e.g. use or not of poly(A)) selection, effects of Actinomycin D). The mRNAs showing clear discrepancies across all datasets could also be interesting, suggesting other types of regulation.

3) The authors must relate their results to the two other ribosome profiling datasets, which are more robust because they have more replicates: Jensen et al. 2014 and Antwi et al.

4) Overall the generation of the data and figures need to be more comprehensively described. (see comment 1).

Figure 1C: How does the new geCAI compare to the CAI or tAI? The authors should provide p-values of mRNA levels vs CAI and tAI score.

Figure 2: "The values for GFPs were adjusted to allow for an effectively 8-fold lower copy number that the endogenous genes." It is not clear to me what this statement means.

Figure 2—figure supplement 2: This figure does not contain sufficient information. E.g. it is not clear what was measured, how often, what cell lines etc. All raw measurements should be listed, otherwise it is impossible to understand what the authors show in panel B (I assume the left panel is A and the right panel B).

Subsection “Gene expression codon adaptation index, geCAI, a codon usage statistic to predictmRNA levels” and Materials and methods section: It is not clear if the RNAseq data was generated as part of the study or previously, as part of the study by Kelly et al., 2017, as that study contains a link to the same dataset.

If the RNA-seq data was generated as part of this study, more details on library preparation should be included, even if it was carried out by BGI. Also, it should be stated how many reads where obtained from the different replicates, what percentage could be mapped, and which genome was used for the alignment, 927 or 427.

[Editors' note: the authors’ plan for revisions was approved and the authors made a formal revised submission. Further revisions were then requested prior to acceptance, as described below.]

Thank you for submitting your article "Codon choice directs constitutive mRNA levels in trypanosomes" for consideration by *eLife*. Your article has been reviewed by three peer reviewers, and the evaluation has been overseen by a Reviewing Editor and Wendy Garrett as the Senior Editor. The following individual involved in review of your submission has agreed to reveal her identity: Christine Clayton (Reviewer #1).

The reviewers acknowledge the work and analyses done to improve clarity and thinking processes. They have discussed the reviews with one another and identified the points that still deserve the authors' attention. the Reviewing Editor has drafted this decision to help you prepare a revised submission. We hope you will be able to submit the revised version shortly.

Summary:

This paper shows that in trypanosomes, as in yeast and metazoa, codon usage affects mRNA stability. The authors used mRNA expression levels calculated from one RNAseq dataset to determine geCAI values for each codon and then tested the ability of these geCAI values to determine the protein and mRNA abundance of GFP transgenes with different codon usage. The protein and mRNA expression levels of these GFP transgenes were consistent with the predictions using geCAI.

Essential revisions:

1) In regard to the 8-fold lower copy of different GFP verses endogenous genes. The response by the authors does not make sense - there should be 4 copies of GFP compared to 8 copies of all single copy diploid genes, especially if there is contamination of one emission spectra with the others.

2) Materials and methods section: In the rebuttal letter, the authors go through great length outlining how differences among RNA-seq datasets may be due to technical reasons. Yet they do not provide much detail on how the RNA-seq libraries were generated for this study.

The first step to making RNA-seq datasets more comparable would be a very detailed description of how the libraries were generated. The authors should provide information on the variables they themselves list:

- The amount of time taken to wash the cells between culture flask and cell lysis; a 1, 5 or 10 minute centrifugation will have a differential effect on mRNA abundance of different genes depending on the half-life of that mRNA.

- The composition and temperature of any wash solution.

- The method used to prepare mRNA.

- The method used for library construction and RNAseq.

- The analysis of the RNAseq data to quantify mRNA expression levels.

In addition, information should be added on the polyA-enrichment step: e.g. what poly-dT beads were used? Magnetic or cellulose? From what manufacturer?

---

## [Author Response]

Summary:This paper shows that in trypanosomes, as in yeast and metazoa, codon usage affects mRNA stability. The authors do this by measuring the abundance and stabilities of variously-coded GFP mRNAs in procyclic trypanosomes. They then use the information, combined with an RNASeq dataset, to calculate optimised scores for the effects of different codons. Importantly, the authors show that translation is required for the codon effects.

This is not quite right. We used mRNA expression levels calculated from one RNAseq dataset to determine geCAI values for each codon and then tested the ability of these geCAI values to determine the protein and mRNA abundance of GFP transgenes with different codon usage. The protein and mRNA expression levels of these GFP transgenes were consistent with the predictions using geCAI.

However, they do not demonstrate that the hairpins are really working to block translation as anticipated.

See point 8 below.

Essential revisions:1) The authors need to improve the clarity of the paper in every section. Concepts need explaining, and the figures need much more detail in their accompanying legends to be comprehensible.

We have re-written substantial parts of the manuscript including figure legends.

The substantially changed text is in blue in the revised manuscript. In addition, a large number of minor changes have been made.

2) In regard to Figure 1 and Figure 7 as well as the others, the authors must Strengthen/weaken the significance of their analysis by showing that their index works on different RNAseq datasets:The authors have optimized their CAI for one dataset (developmentally un-regulated genes and tested for mRNA from PF cells). The authors state that their dataset differs from all the others and therefore comparisons are meaningless.

We did not mean to give the impression that the comparison with other datasets is meaningless. We wrote ‘The ability to compare geCAI with other datasets was restricted by the low correlation values between RNA-seq measured mRNA levels determined in different labs for PCFs (Supplementary file 9A)’.

What we meant by this is that the calculation of the values for geCAI are dependent on measured transcriptome mRNA abundances and these vary in datasets from different labs (see below). So, the use of the geCAI values derived from the transcriptome measured here and then applied to another dataset (with different values for mRNA abundances) would not be the best approach. The best route is a calculation of geCAI values for each of the other datasets to compensate for the variation in measured mRNA levels.

The RNAseq quantitation of transcriptomes vary for the reasons outlined below and it is likely that all, including the dataset we produced, represent approximations to the actual mRNA levels in cells. This variation is not unique to trypanosome labs and has been described in more detail in yeast. The important point here is not that one set of values for geCAI allows predictions for all datasets but that the calculation of geCAI for a dataset allows prediction of expression for mRNAs measured using the same conditions for growth, RNA prep etc. This calculation is necessary to compensate for the variations in measures of mRNA abundance.

Why is there variation between the RNAseq derived mRNA expression levels in data from different labs? There are two non-exclusive possibilities described below:

1) The differences represent real differences in mRNA expression level due to different genotypes; different adaptations to culture, different culture conditions including cell density at harvesting and media composition, including cell density at time of mRNA harvesting.

2) The differences arise for technical reasons, bearing in mind that most stable mRNAs have half-lives of 10 to 30 minutes:

- The amount of time taken to wash the cells between culture flask and cell lysis, a 1, 5 or 10 minute centrifugation will have a differential effect on mRNA abundance of different genes depending on the half-life of that mRNA.

- The composition and temperature of any wash solution.

- The method used to prepare mRNA.

- The method used for library construction and RNAseq.

- The analysis of the RNAseq data to quantitate mRNA expression levels.

Looking at the published methods for the available datasets, there are differences in at least 8 of the variables above and this excludes pre-treatment with cycloheximide used in the ribosome foot printing preotocols which is known to alter mRNA levels within <10 minutes (see Figure 2 in Webb et al., 2005 as an example). No two datasets from different labs used identical methodology and the correlation between RNAseq datasets reflected this in by the analysis in Supplementary file 8.

We have calculated geCAI values for other datasets to show that the method for generating geCAI predictions of mRNA levels is applicable to all datasets.

We used the following more recently published RNA seq datasets; the methodology and analysis has vastly improved in the last 5 years, in particular the length of sequence read and these are generally more reliable.

https://www.ncbi.nlm.nih.gov/pubmed/28742275 (Christiano et al., 2017 – PCF data)

https://www.ncbi.nlm.nih.gov/pubmed/27756224 (Hutchinson et al., 2016 – BSF and PCF)

https://www.ncbi.nlm.nih.gov/pubmed/25145465 (Fadda et al., 2014 – PCF data as also used for PCF half-lives (see below))

Importantly, we have also provided a script on GitHub so that the calculation of geCAI values is accessible for anyone to use on their own data.

We have also highlighted in the text that the geCAI values are dependent of strain, conditions and lab and computing protocols. We realise now that we should have emphasised this much more strongly in the original manuscript.

Supplementary file 3 gives the correlations between dataset and shows that the best correlation is between two datasets from the same study but from different developmental forms. The fact that this correlation between two developmental forms is better that between datasets for the same developmental form from different studies suggests.

However, the reviewers disagree. If their CAI only works with their own dataset then it is the CAI that is problematic.

See above; the problem arises due to the variation in RNAseq datasets not with the geCAI method. The method will work with all datasets but the geCAI values would have to be recalculated to allow for variations due to the factors listed above.

Also, although they claim that the CAI reflects mRNA half-life, the authors ignore the published transcriptome-wide half-life measurements.Please show the following scatter plots (essential):a) CAI vs other procyclic form RNASeq datasets. (In addition to the datasets mentioned, there are also some procyclic ones in 10.1093/nar/gkv731, and the original Siegel et al. set.)b) CAI vs a few bloodstream form datasets (excluding developmentally regulated mRNAs, of course)c) CAI versus mRNA half-life (Fadda et al., 2014).

This has been done.

The results might be less compelling than those obtained using the dataset with which the CAI was trained, but probably not completely random. Nevertheless, the authors can then discuss the strengths and limitations of their CAI in a balanced way- and also, of course, the limitations of some of the datasets (e.g. use or not of poly(A)) selection, effects of Actinomycin D). The mRNAs showing clear discrepancies across all datasets could also be interesting, suggesting other types of regulation.3) The authors must relate their results to the two other ribosome profiling datasets, which are more robust because they have more replicates: Jensen et al., 2014 and Antwi et al.

The arguments above apply to this point as well. The use of more replicates obviously reduces the error in mRNA abundance measurements for the cell line, growth conditions, RNA extraction method, and RNAseq protocol and analysis that was used. It does not in itself make it a more accurate measure of the absolute values for mRNA abundance for procyclic form *T. brucei* (the actual levels in the cell). As stated above, biological and technical variability between strains and experiments mean that mRNA abundance estimates for the same genes vary between experiments. This is the expected behaviour and not a limitation of our (or any other) study.

We have used one of the additional ribosomal profiling datasets. We did not use the Antwi et al. dataset as values were only available for ~60% of the mRNAs, whereas the other two contained values for >98% of the 5136 genes used for the analysis. The two datasets used gave similar results.

4) Overall the generation of the data and figures need to be more comprehensively described (see comment 1)Figure 1C: How does the new geCAI compare to the CAI or tAI? The authors should provide p-values of mRNA levels vs CAI and tAI score.

This has been added. Values were given for the GFP transgenes in Supplementary File 3B.

Figure 2: "The values for GFPs were adjusted to allow for an effectively 8-fold lower copy number that the endogenous genes." It is not clear to me what this statement means.

This has been expanded in the text. The RNAseq was performed on a mixture of four cell lines each containing a haploid copy of a GFP transgene. Trypanosomes are diploid, so any endogenous single copy gene will be present in two copies per cell in all four cell lines. So, in the mixed population there are eight copies of an endogenous gene for one copy of each of the GFP transgenes.

Figure 2—figure supplement 2: This figure does not contain sufficient information. E.g. it is not clear what was measured, how often, what cell lines etc. All raw measurements should be listed, otherwise it is impossible to understand what the authors show in panel B (I assume the left panel is A and the right panel B).

The figure shows a comparison of GFP quantitation by flow cytometry and by quantitative western blotting. We have improved the figure.

Subsection “Gene expression codon adaptation index, geCAI, a codon usage statistic to predictmRNA levels” and Materials and methods section: It is not clear if the RNAseq data was generated as part of the study or previously, as part of the study by Kelly et al., 2017, as that study contains a link to the same dataset.If the RNA-seq data was generated as part of this study, more details on library preparation should be included, even if it was carried out by BGI. Also, it should be stated how many reads where obtained from the different replicates, what percentage could be mapped, and which genome was used for the alignment, 927 or 427.

The RNAseq data is the same as in the reference, the two manuscripts were written in parallel and the absence of the reference was an oversight. This has now been addressed.

[Editors' note: further revisions were requested prior to acceptance, as described below.]

Essential revisions:1) In regard to the 8-fold lower copy of different GFP verses endogenous genes. The response by the authors does not make sense- there should be 4 copies of GFP compared to 8 copies of all single copy diploid genes, especially if there is contamination of one emission spectra with the others.

It does make sense. As stated in the manuscript: mRNA was extracted from a mixture of equal numbers of cells from four cell lines each expressing a differently coded GFP from a single integrated transgene (haploid). Let’s call them GFPA, GFPB, GFPC and GFPD. If we imagine one hundred cells in the mixture, 25 will be expressing GFPA, 25 will express GFPB and so on. So, in the one hundred diploid cells in the mixture, there will be 200 copies of a normal single copy gene and 25 copies of each GFP transgene, that is one eighth.

We have modified the text to try and clarify further.

I am not sure how one emission spectrum can contaminate another. In addition, all GFPs had exactly the same amino acid sequence and so the same emission spectra. We used GFP ORFs that were different enough to be distinguished with certainty in the RNAseq reads.

2) Materials and methods section: In the rebuttal letter, the authors go through great length outlining how differences among RNA-seq datasets may be due to technical reasons. Yet they do not provide much detail on how the RNA-seq libraries were generated for this study.The first step to making RNA-seq datasets more comparable would be a very detailed description of how the libraries were generated. The authors should provide information on the variables they themselves list:- The amount of time taken to wash the cells between culture flask and cell lysis; a 1, 5 or 10 minute centrifugation will have a differential effect on mRNA abundance of different genes depending on the half-life of that mRNA.

Provided.

- The composition and temperature of any wash solution.

Provided.

- The method used to prepare mRNA.

Provided.

- The method used for library construction and RNAseq.

Provided.

- The analysis of the RNAseq data to quantify mRNA expression levels.

Provided.

In addition, information should be added on the polyA-enrichment step: e.g. what poly-dT beads were used? Magnetic or cellulose? From what manufacturer?

This was performed by BGI as part of the BGI RNAseq procedure described.